# Lifetime stability of social traits in bottlenose dolphins

Taylor Evans [ID] [1✉], Ewa Krzyszczyk [ID] [1], Céline Frère[2] & Janet Mann[1]

Behavioral phenotypic traits or "animal personalities" drive critical evolutionary processes such as fitness, disease and information spread. Yet the *stability* of behavioral traits, essential by definition, has rarely been measured over developmentally significant periods of time, limiting our understanding of how behavioral stability interacts with ontogeny. Based on 32 years of social behavioral data for 179 wild bottlenose dolphins, we show that social traits (associate number, time alone and in large groups) are stable from infancy to late adulthood. Multivariate analysis revealed strong relationships between these stable metrics within individuals, suggesting a complex behavioral syndrome comparable to human extraversion. Maternal effects (particularly vertical social learning) and sex-specific reproductive strategies are likely proximate and ultimate drivers for these patterns. We provide rare empirical evidence to demonstrate the persistence of social behavioral traits over decades in a non-human animal.

[1] Department of Biology, Georgetown University, Washington, DC, USA. [2] Global Change Ecology Research Group, University of the Sunshine Coast, Sippy Downs, QLD, Australia. ✉email: tcc55@georgetown.edu

For most animals, ontogeny involves dramatic changes beyond physiological maturation. Social environments expand and shift with independence, hormones rage and fade, and reproduction brings a new set of challenges. In the midst of this, individuals may display behavioral traits: consistent differences in behaviors across time and/or context. Here, we use the term behavioral trait, but this phenomenon has also been referred to as behavioral phenotypes or animal personality (Table 1). Behavioral traits are common across taxa[1], heritable[2,3], and influence population level dynamics such as niche partitioning[4–6], responses to environmental change[7–9], disease transmission[6,10,11], and fitness[12–14]. Traits can also be organized into correlated suites of behaviors, known as syndromes (Table 1). The degree of plasticity for many behaviors is widely debated, exacerbated by redundant terminology, but the widespread presence of behavioral traits precludes behavior from being considered a completely labile characteristic. The extent to which behavioral traits and their syndromes persist throughout ontogeny is still poorly understood, and no obvious patterns have emerged in the literature. Studies have shown temporal stability on relatively short scales[15], and behavioral traits may or may not last across extreme ontogenetic shifts such as metamorphosis[16–19]. Long-lived, social mammals have protracted dependency periods, extensive behavioral repertoires, and can experience environmental change over multiple years and seasons, providing an opportunity to test the limits of behavioral stability. To this end, we examined the extent of stability in individual social behavior across decades, from infancy to late adulthood, in the Indo-Pacific bottlenose dolphin (*Tursiops aduncus*).

Behavioral traits are by definition stable over time, but the extent of temporal stability, either by raw number of years or percentage of an individual's lifespan[20], is not well defined. Most studies encompass only 1 year's worth of testing[21], and studies that do report data across years often cover only a minor portion of their subjects lifetime[13,22–25]. When behavioral stability has been measured over the majority of an animal's lifetime, the focus is on species with short lifespans[15,26–28]; such species typically experience little ecological, demographic, or social variation. Understanding the role of ontogeny is essential for identifying both causal mechanisms and age-specific selective pressures that lead to the maintenance of behavioral traits and syndromes[29,30]. A handful of studies have found that behavioral traits are stable through early maturation[23,28,31–33], hinting that behavior does persist across substantial ontogenetic change. But the majority of studies still include only one developmental stage, most often adult,[34] consigning us to a limited understanding of behavioral stability through ontogeny. By expanding the time span and developmental stages studied, we can better understand how behavioral traits are formed and maintained. Longitudinal studies of long-lived species require substantial human investment and resources, making datasets on behavioral traits across their life history stages quite rare.

In general, studies of behavioral traits and syndromes have focused on boldness and exploration, in ecological and/or social settings[1,35]. Social traits, typically defined by individual variation in gregariousness or social tolerance, have received comparatively less direct attention[21] but are a rich area for study given that individual social style affects ecologically relevant factors such as dispersal[36], mating behavior[37], home-range size, and habitat use[38]. Social traits have also been linked to reproductive success[39], but different types of social connectedness can have differential effects on survival[13,40,41]. How observed social behaviors can be used to define a social trait or syndrome is not simple. Many longitudinal studies of social behavior focus on the stability of dyadic relationships within highly stable groups rather than individual social styles[13,42,43]. A growing number of studies define social traits by network position[44–46], but this obscures *individual* preferences from the social environment. While some network metrics, such as degree, can be indicative of individual-level traits, most are aimed at describing the topology and higher-order properties of the network[47]. Social metrics measured at the individual level better describe individual social decisions, such as whether to join or leave a group. By incorporating several individually measured social behaviors (such as group size preference) and examining their stability across time and correlation among individuals, we can better operationalize a "sociable" behavioral syndrome[48] along the lines of human social personality (e.g., extraversion, agreeableness).

The bottlenose dolphins of Shark Bay, Western Australia, provide a unique opportunity to examine social traits. Their life spans are very long (40+ years[49]) with an extended developmental period (average weaning age is 4 years, and average age at first birth is 13 years)[49–51]. Shark Bay dolphins live in an open fission-fusion society; individual dolphins join and leave groups at will several times per hour, and can associate with any conspecifics they choose within their home range from an unbounded network[52–55]. This flexibility makes group size preferences an informative metric. Societies classified as fission-fusion often retain structures or hierarchies which influence group membership, so associations are made based on factors other than individual social preference (such as rank)[56]. Because Shark Bay bottlenose dolphins live in a large, open fission-fusion system with no clear social hierarchy, individual social decisions are not driven by rigid social structures; consequently individual traits can be better isolated. Additionally, social behavior in this

**Table 1 Definitions for levels of behavioral measurements, as used in this paper.**

| Term | Definition | References |
|------|-----------|-----------|
| Behavioral trait | Differences at the between-individual level in a single behavior that are consistent (repeatable) across time and/or context. Often termed 'animal personality' in the literature. | Sih et al.[1] |
| Behavioral syndrome | A suite of behavioral traits that are correlated at the between-individual level. Use of a multivariate approach allows partitioning of between-individual correlations from phenotypic and among-individual correlations. | Sih et al.[1], Thys et al.[99] |
| Personality | The endogenous (but not immune to environmental influences) temperament or disposition of an individual, or the individual's characteristic patterns of thoughts, feelings, and actions. Personality is largely measured through self and peer ratings, and is most analogous to behavioral syndromes. Here personality is used exclusively to describe human-focused studies. | McCrae et al.[100], Lee & Ashton[101], McCrae & Costa[102] |
| Behavioral phenotypes | An individually expressed behavior or behavioral strategy, not necessarily repeatable (e.g., alternative mating strategies such as singer or satellite males). | Dominey[103] |

population is both heterogeneous across individuals and tied to fitness[41,53]. Mating strategies drive male social behavior, with males forming long-term same-sex alliances to secure temporary access to cycling females[57]. Foraging ecology appears to influence female social behavior, varying from solitary to gregarious depending on the time demands of individual foraging tactics[58–60]. Equipped with 32 years of longitudinal data, we measured the stability (i.e. repeatability, or proportion of variation due to the individual[61,62]) of seven social measurements across maturation from calf to adulthood and into old age. We quantified time spent alone, time in small groups, time in large groups, raw number of associates, same-sex associates, time in socially active groups, and time foraging (individuals in this population forage alone so time foraging is a proxy for nonsocial activity budget) for nine age blocks. This allowed us to observe mean-level changes in these social behaviors across the lifespan. Then repeatability was calculated for each measurement. This was done for the entire population as well as split by sex to account for sex differences in social behavior. Finally, to address the fact that single measurements may be structured into a broader social syndrome, we tested the correlation between repeatable social measurements using multivariate analyses. By incorporating all ontogenetic stages and an extended time period, we provide a robust framework for understanding the architecture of social traits outside of humans and the selective pressures which could be maintaining them.

## Results

**Repeatability.** Generalized linear mixed models for each of the seven traits measured revealed age- and sex-related patterns in social behavior (Supplementary Fig. 1). In general, males were more gregarious than females, and sociability peaked around the juvenile and early adult period, with a decline in old age (Supplementary Fig. 1). After extracting variance components and calculating repeatability estimates for each trait, four of the seven were highly repeatable for both sexes (time alone, time in large groups, number of associates, and same-sex associates), indicating strong and stable individual traits (Fig. 1). Time in small groups was highly repeatable for males but not females, and time foraging was repeatable for females but not males (Fig. 1). Time in socially active groups was not significantly repeatable at the individual level in any model.

**Correlation between social measurements.** A multivariate model of the highly repeatable traits showed that social traits correlated significantly at the among-individual level, forming a behavioral syndrome (Table 2). When split by sex, the models indicated that the traits that were repeatable for only one sex (time foraging for females, time in small groups for males) were also correlated with this syndrome, although in the male-only model time alone did not significantly correlate with the other traits except for time in large groups (Table 2). Principal components analyses including both sexes revealed that 76% of the variation in the four repeatable measurements could be condensed onto a single axis (Fig. 2), strongly suggesting an underlying "sociable" axis of behavior. All four repeatable measures loaded onto the first principal component (Table 3). When split by sex this axis still explained a substantial amount of variation (86% for females, 61% for males, Table 3). All four of the highly repeatable social measures were significantly correlated at the phenotypic level (p < 0.001, Supplementary Fig. 2), with time alone negatively correlated to the other three and positive correlations between the rest.

## Discussion

This study provides the first empirical evidence to date of social trait stability across decades outside of humans, advancing our understanding of behavioral repeatability. In the Shark Bay dolphins, time spent alone, time spent in large groups, average number of associates, and average number of same-sex associates showed high levels of individual repeatability for both sexes. Time spent foraging was highly repeatable for females but not males, while time in small groups was repeatable for males but not females. Time in socially active groups, presumably a direct measure of sociality, was not repeatable for either sex, indicating that social metrics must be validated as repeatable before assuming they represent an individual trait. Furthermore, repeatable traits formed a strong social behavioral syndrome, emphasizing the correlative nature of social behavior. However, the pattern and magnitude of correlation within the syndrome differed between sexes.

The repeatability values for these dolphins were remarkably high compared to other behavioral trait studies[21]. This is counterintuitive given the dynamic nature of their social system, which lacks a clear kinship[53] and dominance structure[63] and is

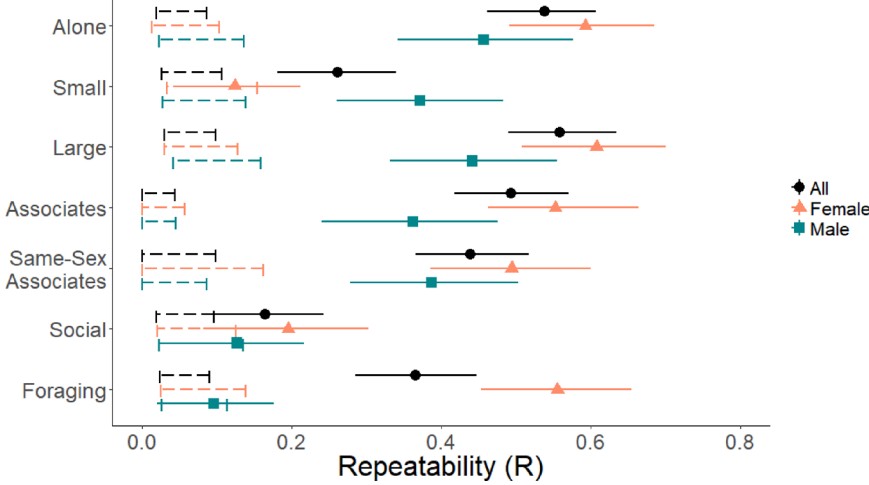

**Fig. 1 Repeatability estimates of social metrics.** Repeatability values for time alone, time in small groups (<6 dolphins), in large groups (≥6 dolphins), average number of associates and same-sex associates, and proportion of sightings in socially active groups and foraging. Values are shown for the entire cohort as well as split by sex. Dashed bars represent the expected repeatability from the null model. Repeatability was considered significant if the 95% credible intervals did not overlap the expected null range. N = 179 dolphins, 89 female, 90 male.

**Table 2 Among-individual correlations for repeatable social measurements.**

**(A) All dolphins**

|  | Alone | Large | Associates | Same-Sex Associates |
|---|---|---|---|---|
| Alone | – | | | |
| Large | **−0.84** (−0.89, −0.77) | – | | |
| Associates | **−0.52** (−0.66, −0.37) | **0.69** (0.59, 0.79) | – | |
| Same-sex associates | **−0.49** (−0.64, −0.33) | **0.71** (0.60, 0.81) | **0.85** (0.80, 0.91) | – |

**(B) Females**

|  | Alone | Large | Associates | Same-sex associates |
|---|---|---|---|---|
| Alone | – | | | |
| Large | **−0.91** (−0.96, −0.87) | – | | |
| Associates | **−0.74** (−0.85, −0.62) | **0.74** (0.63, 0.85) | – | |
| Same-sex associates | **−0.85** (−0.93, −0.77) | **0.87** (0.79, 0.93) | **0.84** (0.75, 0.91) | – |
| Foraging | **0.82** (0.73, 0.91) | **−0.85** (−0.92, −0.77) | **−0.62** (−0.78, −0.44) | **−0.76** (−0.87, −0.62) |

**(C) Males**

|  | Alone | Large | Associates | Same-sex associates |
|---|---|---|---|---|
| Alone | – | | | |
| Large | **−0.63** (−0.80, −0.46) | – | | |
| Associates | −0.02 (−0.27, 0.29) | **0.52** (0.30, 0.72) | – | |
| Same-sex associates | 0.14 (−0.14, 0.40) | **0.36** (0.12, 0.62) | **0.85** (0.77, 0.92) | – |
| Small groups | 0.01 (−0.26, 0.30) | **−0.64** (−0.81, −0.47) | **−0.76** (−0.89, −0.62) | **−0.67** (−0.85, −0.50) |

95% credible intervals for each estimate are in italics. Bolded numbers indicate significant correlation, as determined by CIs that do not cross zero. For the female and male-specific analyses, the trait which was only repeatable for that sex was included.
N = 179 dolphins, 89 female, 90 male.

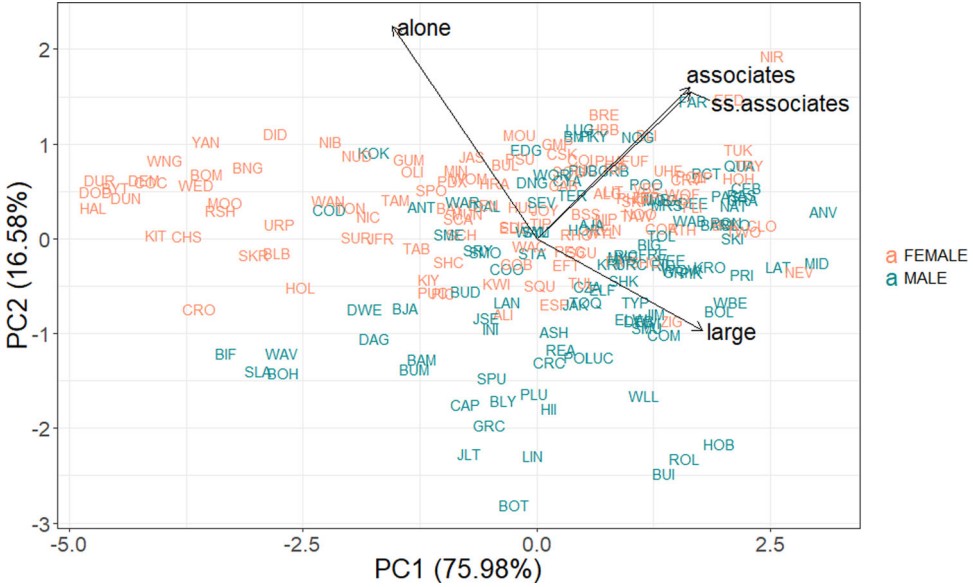

**Fig. 2 Biplot of individual PCA scores.** PCA included only metrics which were significantly repeatable (time alone, in large groups ≥6 dolphins, average number of associates and same-sex associates). Females exhibited more variation along PC1, the axis explaining most of the observed variation. N = 218 dolphins, 112 female, 106 male.

| | PC1: All | PC1: Female | PC1: Male |
|---|---|---|---|
| Alone | −0.47 | −0.48 | −0.25 |
| Large | 0.53 | 0.51 | 0.55 |
| Associates | 0.50 | 0.50 | 0.59 |
| Same-sex associates | 0.50 | 0.50 | 0.53 |
| Total % of variation | 75.98 | 86.10 | 61.18 |

**Table 3 Loadings of the social measurements onto the first principle component, both total population and split by sex.**

All measures were present on PC1, but time alone loaded heavily onto PC2 as well (all = 0.68, females = 0.77, males = 0.80). For all individuals, PC2 explained 16.58% of variation, for females 7.36%, but for males 30.42%.
N = 218 dolphins, 112 female, 106 male.

characterized by extreme fission-fusion dynamics where group size and composition changes >5 times per hour[55]. However, we argue that the very nature of fission-fusion dynamics may in fact allow for individual preferences in social behavior to become more pronounced, resulting in individuals displaying stronger long-term stability in social trait differences. Other studies of fission-fusion societies have noted that individual variation in sociability over shorter time periods can only be partially explained by factors such as home-range overlap and kinship[23,64,65], corroborating this argument. Alternatively, fission-fusion dynamics may require individual stability. Dolphins form long-term social bonds, and are able to recognize each other for decades[66]. Given the hundreds of individual social contacts that dolphins have in their lifetimes, stable social traits might be critical for formation and maintenance of varied relationships. Extreme fission-fusion social dynamics may well depend on stability and predictability of these interactions. In other words, it is necessary to know what to expect from your conspecifics when navigating interactions among flexible group contexts. Stability might be a fundamental feature of understanding sociality and relational complexity[67].

The repeatable social traits we measured correlated to form a social syndrome. This underscores the idea that various social metrics and behaviors stem from the same underlying variation. Too much independence has likely been assigned to different social behaviors, and caution is warranted when multiple measures of sociality are used. To our knowledge, this is the first demonstration of a stable social syndrome, challenging the perception of social behaviors as context-specific[68]. The social behavioral syndrome constitutes a stable framework for the expression of social traits, akin to the extraversion-introversion axis of social expression in humans. Within this stable framework however, group composition, size, and context frequently change, patterns that require some flexibility. Future work could incorporate network dynamics to better parameterize how much social plasticity dolphins might demonstrate.

The behavioral syndrome strength and structure varied by sex. It is well documented that males and females have different pressures across life history stages[69], but we provide new evidence that these sex differences can influence how behavioral traits covary. For females, all repeatable metrics measured, including social and foraging behavior, were very tightly correlated. This is consistent with our previous findings that female social behavior is largely driven by individual foraging strategy, a pattern exemplified by sponge-tool-using females in the population, who forage more and socialize less than other females[60]. For males on the other hand, syndrome correlations were substantial but lower compared to females, and time spent alone did not correlate with the other traits of the syndrome except for time in large groups.

This follows the pattern of male behavior showing lower repeatability values in general. During infancy, males, constrained by maternal behavior and her social network[59], are expected to diverge more from their mothers post-weaning as their life history pressures shift towards alliance formation and reproduction[41]. Male sociality is driven by mating strategy, supported by the addition of time in small groups as a repeatable trait for males (but not females), as alliance membership generally falls in the range of small groups (2–6). Given that alliance membership is likely an arbiter of reproductive success for males, they show less social variation than females, leading to a weaker (but still defined) male behavioral syndrome. In order to successfully navigate and maintain an alliance, males must spend a larger portion of their time in groups, generating a trade-off in time spent foraging (usually a solitary activity). Accordingly, males are less likely to specialize or engage in time-intensive foraging tactics, leading to less variation between males and the resulting lack of repeatability found. The time males do spend alone, which was only correlated (negatively) with time in large groups, could be driven by other aspects of male behavior and within-alliance dynamics.

The lifetime stability of this social syndrome can only be rivaled in timescale by human studies. Human personality (the human-centric version of a behavioral syndrome, Table 1) exhibits temporal stability over long periods[70,71]. However there is evidence that it may be most predictable only in the adult period[72] and that stability may wane over the many decades of a human lifespan[73]. Outside of humans, longitudinal datasets are rare and sex-biased dispersal[74] limits the examination of sex differences in behavioral traits. Other mammalian studies have found stability in sociability, but these patterns are often conflated or intertwined with dominance structure[65,75–77] (but see Seyfarth et al. [13]). Our study is a unique contribution to the non-human literature as it covers all life history stages as well as both sexes.

The strength and duration of social trait repeatability found in the Shark Bay dolphins and its organization into a syndrome offers new insight into the formation and maintenance of behavioral traits. Maternal effects are strongly implicated, given that social dynamics in the calf period were stable well into adulthood. Young animals often inhabit very different niches from adults, and their behavior may differ as a result[30]. However, social traits in the Shark Bay dolphins exhibited stability at the individual level across ontogenetic change. As these patterns are maternally inherited, i.e., calf social traits are strongly influenced by the mother, just as ecological traits are[78,79], mothers play a foundational role in the trajectory of their offspring, especially daughters, but also for sons[41]. It has also been theorized that increased sociality at the species level is a driver of stability in behavioral traits[80], aligning with the extremely high repeatability values found in these very social dolphins. But more comparative studies are needed to parse this theory fully.

We provide rare empirical evidence that social behavioral traits are stable throughout an individual's lifetime, a common but unverified assumption in the literature[35,81–84]. We push the limits of longitudinal studies on behavioral traits, extending number of years measured, percentage of lifespan covered, and length of lifespan involved. Our study suggests that a stable social syndrome shepherds the individual dolphin through substantial physiological and social changes as they move from dependency to adulthood. Dolphins, long suspected of having personality[22,85,86], do indeed show the hallmarks of life-long social traits, perhaps more so than humans.

## Methods

**Study site and population.** This study drew from 67,851 dolphin observations collected through the Shark Bay Dolphin Research Project (SBDRP) between 1988

and 2019. Researchers with the SBDRP have collected behavioral, demographic, genetic, and ecological data on >1800 Indo-Pacific bottlenose dolphins (*Tursiops aduncus*) in Shark Bay, Western Australia since 1984. The study site covers roughly 500 km² in the eastern gulf of Shark Bay, offshore from Monkey Mia (25° 47′S, 113° 43′E). The population is residential and bisexually philopatric[54], allowing data collection to span individual lifetimes for both sexes. Individuals are identified using photo-identification of dorsal fin shape and damage, pigmentation, and other obvious scars (i.e. shark bites or tooth rakes)[87,88]. The sex of each dolphin included in this study was determined by views of the genital area or by association with a calf, and/or with genetics[58,89]. All calves in this study had a known mother based on association. Age was determined by known birth dates[87], size, or degree of ventral speckling[90]. Based on these ages, dolphins were classified as either calves, juveniles, or adults at time of observation. The calf stage was defined as birth until weaning, as determined by the midpoint between last sighting in infant position or at least 80% of time spent with the mother, and when the association between mother and calf declined below 50%[87]. Adult stage was defined as older than 10 years, as 9 is the earliest an individual has become pregnant (an outlier) and the average age of first pregnancy is 13[50], and juvenile as the ages between weaning and adulthood. To account for the extreme length of adulthood compared to calf and juvenile periods, the adult period was split into 5-year blocks from age 10 up until age 50. Only two dolphins had sufficient data past age 50, so this time period was excluded (Supplementary Tables 1, 2).

**Ethics statement**. Research was conducted under Georgetown University Animal Care and Use permits: IACUC-13-069, 07-041, 10-023 and 2016-1235; and Department of Parks and Wildlife Permits (Western Australia): SF-009876, SF-010347, SF-008076, SF009311, and SF007457.

**Behavioral data**. Behavioral data were collected from boat-based observational surveys. Surveys are a 5-min scan sample conducted when dolphins are sighted to determine group composition and predominant group activity (i.e. travel, rest, social, forage, other[91]). Individuals were included in a group according to the 10 m chain rule, where any dolphins within 10 m of one another were considered a single group[58]. To be included in the study, individuals were required to be sighted in at least 15 surveys (as this is where social measurements tend to stabilize[92]) in three consecutive time blocks. If individuals were sighted multiple times in a day, only the last survey in which it was sighted was included in order to reduce spatial and temporal autocorrelation[54]. In order to account for uneven sampling, we drew a random subsample of 15 surveys per individual's life history stage and calculated seven social measurements. This was repeated 1000× to create an average value per measurement for each individual. Social measurements included proportion of surveys alone (including mothers with dependent calves), to capture how often an individual decided to associate with other dolphins or not. Group size preferences when with others were then calculated as time spent in small groups (2–6 individuals, based on the average group size of 6 found in the dataset), and large groups (greater than the average group size of 6). Additionally, we calculated the average number of unique associates (i.e., dolphins sighted with each focal dolphin). This population has high levels of sexual segregation, with females primarily avoiding adult males[55]. This segregation may obscure individual preferences in number of associates when both sexes are counted, therefore we also calculated the average number of same-sex associates. Finally, we calculated the proportion of surveys in socially active groups (where ≥50% of the group is exhibiting social behaviors), as well as proportion of surveys foraging, as this population generally forages alone.

**Repeatability**. In order to maximize sample size, this study utilized an unbalanced design for longitudinal data[93]. The lifespan was divided into nine-time blocks: calf, juvenile, and adulthood, broken into 5-year blocks from 10 years up until age 50 (there were only two dolphins with sufficient surveys past age 50). 179 dolphins met the data requirements (15 or more sightings in at least 3 consecutive time blocks) to be included, 89 females and 90 males. This dataset included 40,523 individual dolphin observations (mean per individual = 377, min 59, max 906). In order to assess the stability of individual social measurements, we measured the repeatability of each social measurement, or the proportion of variation attributed to among-individual difference. This is calculated as the intraclass correlation coefficient (ICC)[21]. Following de Villemereuil et al.[62], we fit generalized linear mixed models with dolphin identity (ID) as a random factor, allowing us to partition the variance due to the individual and estimate the repeatability. The models also included age class and individual sex as fixed factors to account for differences in social behavior present based on these categories. Models were run with a Bayesian framework using the MCMCglmm package[94] in R (version 4.0.2)[95]. The models were assigned a Gaussian error distribution, and all measurements except associates and same-sex associates were square-root transformed to improve normality. Fixed effects were given uninformative priors, and the random effect (dolphin I.D.) was given a weakly informative inverse gamma prior. Models were run for 100,000 iterations with a thinning interval of 10 and a burn-in period of 3000 iterations. Model convergence was checked by visually examining trace plots, autocorrelation, and the effective sample size. Fixed effects were considered significant if the associated credible intervals of the posterior distribution did not cross zero. The ICC for each measurement was then calculated by extracting the variance components of the models and calculating the proportion of variance due to the individual component. Social measurements are inherently non-independent, so it is critical to use a null model to assess the significance of repeatability[44,48]. To accomplish this, we randomized the identity associated with each observation and recalculated the social measurements and their repeatability due to chance. Observed repeatability values were then considered to be high if their 95% credible interval did not overlap the range generated by the null model.

**Correlation of social measurements**. Since different social measurements may be measuring the same underlying variation, we used a multivariate model to examine the correlation of repeatable traits at the among-individual level following Houslay and Wilson[96]. The same model structure as before was used, but with all four repeatable metrics included as a multivariate trait. Models were run for 200,000 iterations with a thinning interval of 100 and a burn-in period of 10,000 iterations. Model convergence was again checked by visually examining trace plots, auto-correlation, and the effective sample size. Posterior distributions of the variance components were then used to assess correlation of the social metrics at the between-individual level. If the 95% credible intervals for a correlation did not cross zero, that correlation was considered significant. This process was done for the population as a whole as well as split by sex.

We also used an exploratory principal components analysis and Kendall's tau correlation to visualize and quantify broader phenotypic-level correlation structure between the highly repeatable measures (Supplementary information). To better characterize large-scale variation and avoid the pitfalls of repeated measurements in principle component analyses, a larger dataset was used for the PCA. Dolphins were included if they met data requirements (15+ surveys) for at least two adult age blocks, and a single adult measurement was then calculated using their adult sightings for the four significantly repeatable social metrics. This avoided variance due to early ontogeny, and increased the sample size for this analysis to better visualize patterns of variance between the sexes. The expanded dataset included 218 dolphins, 112 female, 106 male. Bartlett's test of sphericity indicated that the correlation matrix of the variables was significantly different from the identity matrix ($p < 0.001$), and thus appropriate for dimensionality reduction. Keyser–Meyer–Olkin tests returned an overall measure of sampling adequacy of 0.71 (individual MSAs for each variable ranged from 0.64 to 0.76). PCA scores were calculated using the psych[97] package, and correlations with the corrplot[98] package, both in R[95].

**Reporting summary**. Further information on research design is available in the Nature Research Reporting Summary linked to this article.

## Data availability

Data used in this study are freely available through Open Science Framework. https://doi.org/10.17605/OSF.IO/RSC9T (https://osf.io/rsc9t/).

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

## Acknowledgements

We acknowledge the Malgana peoples, the traditional custodians of Gutharraguda and Irrabuga Mia where this work was conducted. Many thanks to our colleagues and research assistants on the Shark Bay Dolphin Research Project; We are also grateful to the rangers and scientists at the Department of Biodiversity, Conservation and Attractions in W. Australia for logistical support. We give special thanks to Monkey Mia Resort and Royal Automobile Club of Australia for field support. Funding support to J.M. comes from NSF grants #0847922, 0820722, 9753044, 0316800, 0918308, 0941487,1559380, 1755229 and ONR 10230702. T.E. received support from the Animal Behavior Society.

## Author contributions

T.E. and J.M. conceived of the study. T.E. analyzed the data and drafted the manuscript. T.E., E.K., C.F., and J.M. contributed data and assisted in writing the manuscript.

## Competing interests

The authors declare no competing interests.
