## [Peer Review File · Communications Biology]

Reviewers' comments:

Reviewer #1 (Remarks to the Author):

The manuscript entitled "Lifetime stability of social traits in a long-lived marine mammal" is a very interesting article that demonstrates how social traits are stable from infancy to late adulthood in a socially and cognitively complex mammal species (the bottlenose dolphin).

The authors draw on 40,523 individual dolphin observations (5-minute samples) from data collected as part of the Shark Bay Dolphin Research Project in Australia between 1988 and 2019. Most analyses were restricted to 179 individuals with 15 or more sightings in at least 3 consecutive time blocks. The data analyses are adequate (e.g., permutation tests for hypotheses, repeatability measures for each social metric), the findings are very interesting, and the paper is generally well written.

I do, however, have a few concerns about the data analysis. It is possible that some of these doubts are caused by the lack of a few methodological details in the manuscript that would be necessary to thoroughly understand and replicate this study. I give exact examples below.

Introduction

The authors comment in different sentences of the introduction that most studies measuring behavioural stability cover only one year of testing and focus on species with short life spans. Although I fully agree with the authors, some readers may not be aware of this. Hence, it would be useful to cite those studies with several years of data. The following are some studies that come to mind with more than one year of data (some of which are cited in the discussion), such as: 2 years of data in primates (Kulachci et al. 2018), 9 years of data in dolphins (Diaz Lopez, 2020), and 4 years in birds (Dingemanse et al., 2004). I suspect there may be other studies.

Line 70 (and elsewhere). I agree with the use of group size preference as an individually measured social behaviour, but this will not be obvious to most readers. This needs to be explained, given the possible correlation between behavioural activities (e.g. different foraging strategies) and group size.

Line 79. It would be helpful if the authors could explain more explicitly why fission-fusion dynamics might make group size preferences an informative metric. I suggest to explain it and back this up with citations because it will not be intuitive to most readers.

Methods

This section reports valuable ecologically relevant data, linking an considerable amount of behavioural data from the wild to long-term social association data in a socially and cognitively complex mammal species.

Line 233 and following paragraph. The authors refer to different concepts (surveys, groups, and sightings), what is the meaning of sighting for the authors?, could there be different groups within the same sighting? This part may be a bit confusing for those readers who are not familiar with the 5 min surveys and 10m chain definition. I suppose that with the 10 m chain definition, at some points individuals can get separated, leading to the creation of new subgroups. How is this variability taken into account? If a dolphin separates 20 metres from another dolphin for 10 minutes and then returns to its side, would we have two different groups, a solitary dolphin and a pair?

Line 241. I appreciate that the authors drew a random subsample of 15 surveys per individual's life history stage in order to account for uneven sampling. I think this is a very good approach which also limits the possible temporal autocorrelation between surveys.

Line 244. It would be good if the authors could explain the criteria they followed to create the different group categories. In my opinion, and from a behavioural point of view, the formation of a pair or a trio could be quite different from a group of 6 individuals. I understand that the authors

have considered the mean as a reference parameter, why not include group size in the analyses as a quantitative and not a categorical variable?

Line 250. I have been reviewing the data used in this study freely available through Open Science Framework (DOI 10.17605/OSF.IO/RSC9T) and have not been able to find information on number of individuals of the same sex. How was this calculation made? Similarly, it is not clear whether the potential effect of predominant behaviour on the selected social traits was taken into consideration. There is a column from which I assume the behavioural data were extracted (First.Five.Activity), but reading the text it seems that only the proportion of surveys in socially active and foraging groups was calculated. Was the possible effect of resting and travelling activities taken into consideration? If not, why not, what happens if the behaviour is unknown? I think this is an important point, as the possible relationship between certain types of behaviour and the social traits observed should be considered.

Could the authors include, in addition to same-sex associates, the number of same-age associates as a variable? I think that the inclusion of this variable would provide additional information to the study.

Line 293 (and following paragraph). I would like to point out that Principal Component Analysis, or PCA, is an exploratory tool and a dimensionality reduction method often used to reduce the dimensionality of large data sets. The only correlation measure calculated was Spearman's Rho. I suggest using two non-parametric methods to measure the correlation between the variables: Spearman's rho and Kendall's tau. Most of the time, these two measures align closely and lead to the same inferences. However, Spearman's rho is more sensitive to error and discrepancies in the data and Kendall's tau has a lower sensitivity to gross error and a lower asymptotic variance. In this section I would suggest a better explanation of the calculation of the PCA with confirmation of the sampling adequacy of the correlation matrix for the PCA. Such verifications can be done by calculating Bartlett's test of sphericity and the Kaiser-Meyer-Olkin test (Budaev, 2010).

Although this paper is unlikely to challenge behavioural ecologists' thinking about the social lives of animals, this article is very interesting because it provides additional empirical evidence that social behavioural traits are stable over an individual's lifetime in a socially and cognitively complex mammal species. I would like to congratulate the authors for the study and highlight the great effort involved in collecting this vast amount of behavioural data over decades of study of a mammal population.

Dr. Bruno Díaz López

Suggested references:

- Budaev, S. V. (2010). Using principal components and factor analysis in animal behaviour research: Caveats and guidelines. *Ethology*, 116(5), 472e480.
- Díaz López, B. When personality matters: personality and social structure in wild bottlenose dolphins, *Tursiops truncatus*. *Animal Behaviour*, 163, 73–84.
- Dingemanse, N.J., Both, C., Drent, P.J., & Tinbergen, J.M. (2004). Fitness consequences of avian personalities in a fluctuating environment. *Proceedings of the Royal Society B: Biological Sciences*, 271, 847–852.
- Kulahci, I. G., Ghazanfar, A. A., & Rubenstein, D. I. (2018). Consistent individual variation across interaction networks indicates social personalities in lemurs. *Animal Behaviour*, 136, 217–226.

Reviewer #2 (Remarks to the Author):

Reviewer comments for Lifetime stability of social traits in a long-lived marine mammal

Evans et al, is that rare manuscript that is both complex yet infinitely readable. The authors should be commended for their writing. This paper is a compelling investigation of dolphin behavioral syndromes using a long-term data set. The best analogy would be a sociability scale in humans.

The authors claim that they are the first to identify these stable traits on the scale of decades and, given my understanding of the prevailing work in this field, I have no immediate reason to doubt that claim. A basic search for comparable studies in elephants only turned up short-term data, nothing on the scale of this paper. This is a very hard thing to study and outside of captivity only the Shark Bay Dolphin Research Project or Sarasota Dolphin Research Project could potentially put forth such data (based solely on possessing long-term data sets in a general sense). This does confirm my anecdotal impressions from animals under human care, however, the authors do the hard work of establishing behavioral syndromes in a population still experiencing natural selection, which makes this paper very important. I believe this is of value broadly to behaviorists, psychologists, biologists, biological anthropologists and the public at large.

Evaluating the evidence, it seems their conclusions are sound given their innovative analysis. They have used appropriate statistics and techniques to overcome the risk of auto-correlation and they report their statistics appropriately, to my reading, given their modeling approach. Given the availability of their data set I'm confident others could perform appropriate analyses, but as previously mentioned only a few entities could replicate this study with different animals (specifically *Tursiops truncatus*).

One question I have stems from the finding that time foraging was repeatable for females but not males. I saw a positive case in the discussion for why females are stable here, but is it the author's contention that the lack of tool use in males explains this lack of inter-individual stability for that sex? Is there something about the male's life history or physiology that explains the inconsistency here. I would just like the authors to speak more to this. My other question relates to lines 169-171, here I am a bit confused as I am reading this in context with lines 120-124 in the results which seem to speak against that. My interpretation is that this is a situation of the male-only model (lines 114-118), however, from the outside this is slightly hard to interpret and some more time with this might help the reader parse through this a bit.

Reviewers' comments:

Reviewer #1 (Remarks to the Author):

The manuscript entitled "Lifetime stability of social traits in a long-lived marine mammal" is a very interesting article that demonstrates how social traits are stable from infancy to late adulthood in a socially and cognitively complex mammal species (the bottlenose dolphin).

The authors draw on 40,523 individual dolphin observations (5-minute samples) from data collected as part of the Shark Bay Dolphin Research Project in Australia between 1988 and 2019. Most analyses were restricted to 179 individuals with 15 or more sightings in at least 3 consecutive time blocks. The data analyses are adequate (e.g., permutation tests for hypotheses, repeatability measures for each social metric), the findings are very interesting, and the paper is generally well written.

I do, however, have a few concerns about the data analysis. It is possible that some of these doubts are caused by the lack of a few methodological details in the manuscript that would be necessary to thoroughly understand and replicate this study. I give exact examples below.

Introduction

The authors comment in different sentences of the introduction that most studies measuring behavioural stability cover only one year of testing and focus on species with short life spans. Although I fully agree with the authors, some readers may not be aware of this. Hence, it would be useful to cite those studies with several years of data. The following are some studies that come to mind with more than one year of data (some of which are cited in the discussion), such as: 2 years of data in primates (Kulachci et al. 2018), 9 years of data in dolphins (Diaz Lopez, 2020), and 4 years in birds (Dingemanse et al., 2004). I suspect there may be other studies.

- These references have been incorporated to better demonstrate the current state of long-term research on behavioral stability, as well as an additional two studies on primates and hyenas. *Lines 46-47*

Line 70 (and elsewhere). I agree with the use of group size preference as an individually measured social behaviour, but this will not be obvious to most readers. This needs to be explained, given the possible correlation between behavioural activities (e.g. different foraging strategies) and group size.

- We have further explained the difference in use of individual vs network metrics, including a citation. *Lines 70-73*

Line 79. It would be helpful if the authors could explain more explicitly why fission-fusion dynamics might make group size preferences an informative metric. I suggest to explain it and back this up with citations because it will not be intuitive to most readers.

- We have added more detail on the open nature of the Shark Bay fission-fusion dynamics, and added a citation to differentiate it from societies where social decisions are based on structures such as dominance or kinship. *Lines 84-86*

Methods

This section reports valuable ecologically relevant data, linking an considerable amount of behavioural data from the wild to long-term social association data in a socially and cognitively complex mammal species.

Line 233 and following paragraph. The authors refer to different concepts (surveys, groups, and sightings), what is the meaning of sighting for the authors?, could there be different groups within the same sighting? This part may be a bit confusing for those readers who are not familiar with the 5 min surveys and 10m chain definition. I suppose that with the 10 m chain definition, at some points individuals can get separated, leading to the creation of new subgroups. How is this variability taken into account? If a dolphin separates 20 metres from another dolphin for 10 minutes and then returns to its side, would we have two different groups, a solitary dolphin and a pair?

- We have cleaned up the language to prevent sighting and survey being used interchangeably. *Lines 251-252* Sighting is used to literally mean that a dolphin was seen, whereas the term survey refers to a group sighting (which might entail one or more individuals). A single survey does not contain multiple groups; if multiple groups are sighted, they will be surveyed sequentially. Surveys that last for longer than 5 minutes tend to incorporate more joins and leaves, but to better standardize the data for this study we used only the dolphins present in the first 5 minute period of a survey and the resulting predominant activity. If a dolphin leaves at some point during the 5 minute sample it is still recorded as seen. In other words, surveys are treated more like snapshots rather than a way to look at changes in group composition over time.

Line 241. I appreciate that the authors drew a random subsample of 15 surveys per individual's life history stage in order to account for uneven sampling. I think this is a very good approach which also limits the possible temporal autocorrelation between surveys.

- Thank you! We hope this approach becomes more widely adopted, especially for longitudinal studies where effort is often uneven by necessity.

Line 244. It would be good if the authors could explain the criteria they followed to create the different group categories. In my opinion, and from a behavioural point of view, the formation of a pair or a trio could be quite different from a group of 6 individuals. I understand that the authors have considered the mean as a reference parameter, why not include group size in the analyses as a quantitative and not a categorical variable?

- More detail has been added on how we chose group categories. *Lines 259-261* We began with the idea of looking at how much time individuals spent alone (vs with others), in order to capture the decision to socialize or not. This was captured in the metric of

time spent alone. After that we wanted to include more detail on group size preferences, and used the mean group size of our dataset to classify groups as 'small' or 'large.' Interestingly, the mean we found is similar to that of other dolphin populations, which we took as a sign that this may be a meaningful break. However, it did not seem practical to differentiate group sizes further, especially given the blocked nature of our time series. If we looked at each group size separately, the number of sightings for each group size would be much smaller, greatly increasing the error in calculated rates of time spent in each group size. And with our study design, treating group size as a quantitative variable would end up as the mean group size per time block, which would be skewed by time spent alone, thus losing some of the detail in how an individual's time budget is spent in regards to grouping. As seen in Galezo et al. 2018, the most common group size is 1.

- o Galezo, A.A., Krzyszczyk, E. and Mann, J., 2018. Sexual segregation in Indo-Pacific bottlenose dolphins is driven by female avoidance of males. *Behavioral Ecology*, 29(2), pp.377-386.

Line 250. I have been reviewing the data used in this study freely available through Open Science Framework (DOI 10.17605/[OSF.IO/RSC9T](https://doi.org/10.17605/OSF.IO/RSC9T)) and have not been able to find information on number of individuals of the same sex. How was this calculation made? Similarly, it is not clear whether the potential effect of predominant behaviour on the selected social traits was taken into consideration. There is a column from which I assume the behavioural data were extracted (First.Five.Activity), but reading the text it seems that only the proportion of surveys in socially active and foraging groups was calculated. Was the possible effect of resting and travelling activities taken into consideration? If not, why not, what happens if the behaviour is unknown? I think this is an important point, as the possible relationship between certain types of behaviour and the social traits observed should be considered.

- The wiki for this study's data has been updated with detailed explanations for each variable. The data table as-is includes all dolphins seen during data collection, not just those with enough data to be included in further analyses. To calculate same-sex associates, you can use the SurveyNumber variable to identify all dolphins present in each survey. From there you can create a table of SurveyNumber, DolphinID, and Sex. Then you can draw your sample of 15 surveys per dolphin included in the study (i.e. those with sufficient data), and use the SurveyNumber to cross-reference which dolphins were present in each subsampled survey and their sex.
- We did not calculate the repeatability of resting, travel, or 'other' behavior, as these are not necessarily reflective of individual social traits. The other social metrics (time alone, time in small groups, time in large groups, and average # of associates) were calculated across all activity states, as repeatability should encompass stability across time and context. When we did look at surveys marked as social, we did not find repeatability, further indicating that predominant activity may not be indicative of individual traits as the categories are so broad. Foraging however, is generally conducted alone so we wanted to use this as another metric for an individual's non-social activity budget, especially given the previously known connection between tool-use and increased solitary behavior.

Could the authors include, in addition to same-sex associates, the number of same-age

associates as a variable? I think that the inclusion of this variable would provide additional information to the study.

- Association with similar-aged individuals in this population is largely driven by availability rather than individual preference. This is especially true for males, as their alliances are formed by age cohort rather than by kinship (Gerber et al. 2020). Given this, we did not think restricting to same-age associates would indicate an individual trait. Age-homophily changes by life-history stage. That is some of our previous work does indicates low age-homophily for calves (Gibson and Mann 2008), but age- and sex-homophily in the juvenile stage (Galezo et al. 2020; Krzyszczyk et al. 2017). Additionally, it is not clear how to determine how close two dolphins need to be in age to be considered same-age. Male alliance members, for example, can vary in age by a few years.
 - Frere, C.H., Krützen, M., Mann, J., Watson-Capps, J.J., Tsai, Y.J., Patterson, E.M., Connor, R., Bejder, L. and Sherwin, W.B., 2010. Home range overlap, matrilineal and biparental kinship drive female associations in bottlenose dolphins. *Animal Behaviour*, 80(3), pp.481-486.
 - Gerber, L., Connor, R.C., King, S.L., Allen, S.J., Wittwer, S., Bizzozzero, M.R., Friedman, W.R., Kalberer, S., Sherwin, W.B., Wild, S. and Willems, E.P., 2020. Affiliation history and age similarity predict alliance formation in adult male bottlenose dolphins. *Behavioral Ecology*, 31(2), pp.361-370.
 - Gibson, Q.A. and Mann, J., 2008. Early social development in wild bottlenose dolphins: sex differences, individual variation and maternal influence. *Animal Behaviour*, 76(2), pp.375-387.
 - Galezo, A.A., Foroughirad, V., Krzyszczyk, E., Frère, C.H. and Mann, J., 2020. Juvenile social dynamics reflect adult reproductive strategies in bottlenose dolphins. *Behavioral Ecology*, 31(5), pp.1159-1171.
 - Krzyszczyk, E., Patterson, E.M., Stanton, M.A. and Mann, J., 2017. The transition to independence: sex differences in social and behavioural development of wild bottlenose dolphins. *Animal Behaviour*, 129, pp.43-59.

Line 293 (and following paragraph). I would like to point out that Principal Component Analysis, or PCA, is an exploratory tool and a dimensionality reduction method often used to reduce the dimensionality of large data sets. The only correlation measure calculated was Spearman's Rho. I suggest using two non-parametric methods to measure the correlation between the variables: Spearman's rho and Kendall's tau. Most of the time, these two measures align closely and lead to the same inferences. However, Spearman's rho is more sensitive to error and discrepancies in the data and Kendall's tau has a lower sensitivity to gross error and a lower asymptotic variance. In this section I would suggest a better explanation of the calculation of the PCA with confirmation of the sampling adequacy of the correlation matrix for the PCA. Such verifications can be done by calculating Bartlett's test of sphericity and the Kaiser-Meyer-Olkin test (Budaev, 2010).

- This is a good point, we have corrected the language in the paragraph beginning on line 309. We have also switched the correlation calculations to Kendall's tau, reflected in the supplementary info. We also reported on Bartlett's test of sphericity and Kaiser-Meyer-Olkin tests, which indicated that the data was suitable for PCA. *Lines 319-324, updated figure from supplementary info below*

- S2. Phenotypic correlations (Kendall's tau) of the four repeatable social measurements (time alone, in large groups ≥ 7 dolphins, average number of associates and same-sex associates). All correlations were significant ($p < 0.001$).

Although this paper is unlikely to challenge behavioural ecologists' thinking about the social lives of animals, this article is very interesting because it provides additional empirical evidence that social behavioural traits are stable over an individual's lifetime in a socially and cognitively complex mammal species. I would like to congratulate the authors for the study and highlight the great effort involved in collecting this vast amount of behavioural data over decades of study of a mammal population.

Dr. Bruno Díaz López

Suggested references:

- Budaev, S. V. (2010). Using principal components and factor analysis in animal behaviour research: Caveats and guidelines. *Ethology*, 116(5), 472e480.
- Díaz López, B. When personality matters: personality and social structure in wild bottlenose dolphins, *Tursiops truncatus*. *Animal Behaviour*, 163, 73–84.
- Dingemanse, N.J., Both, C., Drent, P.J., & Tinbergen, J.M. (2004). Fitness consequences of avian personalities in a fluctuating environment. *Proceedings of the Royal Society B: Biological Sciences*, 271, 847–852.
- Kulahci, I. G., Ghazanfar, A. A., & Rubenstein, D. I. (2018). Consistent individual variation across interaction networks indicates social personalities in lemurs. *Animal Behaviour*, 136, 217-226.

Reviewer #2 (Remarks to the Author):

Reviewer comments for Lifetime stability of social traits in a long-lived marine mammal

Evans et al, is that rare manuscript that is both complex yet infinitely readable. The authors should be commended for their writing. This paper is a compelling investigation of dolphin behavioral syndromes using a long-term data set. The best analogy would be a sociability scale in humans. The authors claim that they are the first to identify these stable traits on the scale of decades and, given my understanding of the prevailing work in this field, I have no immediate reason to doubt that claim. A basic search for comparable studies in elephants only turned up short-term data, nothing on the scale of this paper. This is a very hard thing to study and outside of captivity only the Shark Bay Dolphin Research Project or Sarasota Dolphin Research Project could potentially put forth such data (based solely on possessing long-term data sets in a general sense). This does confirm my anecdotal impressions from animals under human care, however, the authors do the hard work of establishing behavioral syndromes in a population still experiencing natural selection, which makes this paper very important. I believe this is of value broadly to behaviorists, psychologists, biologists, biological anthropologists and the public at large.

Evaluating the evidence, it seems their conclusions are sound given their innovative analysis. They have used appropriate statistics and techniques to overcome the risk of auto-correlation and they report their statistics appropriately, to my reading, given their modeling approach. Given the availability of their data set I'm confident others could perform appropriate analyses, but as previously mentioned only a few entities could replicate this study with different animals (specifically *Tursiops truncatus*).

One question I have stems from the finding that time foraging was repeatable for females but not males. I saw a positive case in the discussion for why females are stable here, but is it the author's contention that the lack of tool use in males explains this lack of inter-individual stability for that sex? Is there something about the male's life history or physiology that explains the inconsistency here. I would just like the authors to speak more to this.

- We are attributing the lack of repeatability in time foraging to the alliance structure typical of adult males. Maintaining their alliance bonds requires more time spent in groups and focus on social dynamics, at the expense of perfecting time-intensive foraging strategies such as tool-use. So between males there is less variation in how much time they spend foraging, as they tend to use tactics common to all members of the population. This social constraint on foraging variation leaves less room for individual trait expression, thus the lack of individual repeatability. We have added some clarification to make sure this is more directly addressed in the manuscript. *Lines 186-190*

My other question relates to lines 169-171, here I am a bit confused as I am reading this in context with lines 120-124 in the results which seem to speak against that. My interpretation is that this is a situation of the male-only model (lines 114-118), however, from the outside this is

slightly hard to interpret and some more time with this might help the reader parse through this a bit.

- Your interpretation is correct, the lower correlations are for the male-only model. There were still substantial correlations for males, but they were lower than what was seen in the female-only model. We have specified that this line is comparing the males to females, and that there was still a syndrome found for males, just not as strongly as for females. *Line 177*

REVIEWERS' COMMENTS:

Reviewer #1 (Remarks to the Author):

In this second revision the authors have followed the advice given to improve the manuscript. I consider that the study is now ready for publication and I can only congratulate the authors on their work.

Dr. Bruno Díaz López